# Optimization of Quality Properties of Gluten-Free Bread by a Mixture Design of Xanthan, Guar, and Hydroxypropyl Methyl Cellulose Gums

**DOI:** 10.3390/foods8050156

**Published:** 2019-05-10

**Authors:** Christian R. Encina-Zelada, Vasco Cadavez, José A. Teixeira, Ursula Gonzales-Barron

**Affiliations:** 1CIMO Mountain Research Center, School of Agriculture, Polytechnic Institute of Bragança, Santa Apolónia Campus, 5300-253 Bragança, Portugal; vcadavez@ipb.pt; 2Center of Biological Engineering, School of Engineering, University of Minho, Gualtar Campus, 4704-553 Braga, Portugal; jateixeira@deb.uminho.pt; 3Department of Food Technology, Faculty of Food Industries, National Agricultural University La Molina, Avenue La Molina s/n, Lima 12, Lima 15024, Peru

**Keywords:** quinoa, rice, maize, texture profile analysis, hydrocolloids, response surface, crumb image analysis

## Abstract

The objective of this study was to investigate, by means of a D-optimal mixture design, the combined effects of hydroxypropyl methyl cellulose (HPMC), xanthan (XG), and guar (GG) gums on physicochemical, rheological, and textural properties of gluten-free batter and bread. For each of the quality properties measured, a two-factor interaction model was fitted, and the significance of its terms was assessed by analysis of variance. Sticky batters were produced with a combination of high dose of GG (0.60%), high-intermediate dose of HPMC (3.36%), and low dose of XG (0.04%). Combinations of high XG dose (0.60%) and intermediate doses of HPMC (3.08%) and GG (0.32%) rendered GF breads of greater specific volume, while lower bread crust luminosity was obtained with combinations of high GG dose (0.60%), low XG dose (0.04%), and high-intermediate HPMC dose (3.36%). Combinations of high-intermediate HPMC dose (3.36%), high GG dose (0.60%), and low XG dose (0.04%) produced both softer crumbs and bread slices of more open visual texture. By using a desirability function that maximized specific volume while minimizing crust luminosity, crumb hardness, and mean cell density, the optimization of hydrocolloids mixture rendered a value of 0.54, for a combination of 0.24% XG, 0.60% GG, and 3.16% HPMC.

## 1. Introduction

Hydrocolloids have been widely used in food products to modify texture, improve moisture retention, control water mobility, and maintain overall product quality during storage [1]. They are frequently used in gluten-free (GF) breadmaking to mimic the visco-elastic properties of gluten, thereby increasing gas retention during proofing and baking; and hence enhancing loaf specific volume [2]. Not only hydrocolloids are frequently used in the formulation of GF breads—such as hydroxypropyl methyl cellulose, carboxymethyl cellulose, xanthan gum, pectin, guar gum, gum arabic, locust bean gum, or carrageenans—but also emulsifiers for instance sodium stearoyl-2-lactylate, diacetyltartaric, and fatty acid esters of glycerol (DATEM) or white egg [3,4,5]. Furthermore, supplementation of common GF flours (rice and maize) with cassava flour, corn starch, and starch from tubers such as potato and tapioca are most commonly used in the GF breadmaking [3,5,6].

Xanthan gum (XG), a polysaccharide produced by the *Xanthomonas campestris* bacterium, is also used in GF breadmaking, since it contributes to obtaining stable foams based on protein–gum interaction, with rheological properties typical of viscoelastic solids [7]. In bakery, guar gum (GG), another polysaccharide obtained from *Cyamopsis tetragonoloba* beans, is used to improve mixing and batter tolerance; to extend the shelf life of products through moisture retention; and to improve loaf volume, bread crumb/crust, and color [8,9,10]. Hydroxypropyl methyl cellulose (HPMC), a non-ionic modified cellulose, is one of the most important hydrocolloids used in the elaboration of GF breads due its capacity to increase batter consistency and spread ratio, bread specific volume, cohesiveness, and porosity of crumb; while decreasing crumb hardness [11,12]. Since each of the hydrocolloids mentioned above provide specific quality attributes to GF batter and bread, it is desirable to evaluate their combined effects and whether they have synergistic effects.

Understanding the effect of combinations of ingredients in food products can be achieved through a special case of the response surface statistical approach: the mixture design [13]. In mixture design experiments, the factors are different components of a blend, and the measured response is assumed to depend only on the relative proportions of the ingredients or components in the mixture and not on the amount of the mixture [14]. Approaches such as D-optimal designs became popular because they are intended to ‘optimally’ cover any type of experimental region [15]. Thus, the objectives of this study were: (i) to investigate, by means of a D-optimal mixture design, the combined effects of HPMC, XG, and GG on physicochemical, rheological, and visual textural properties of gluten-free batter and bread; and (ii) to optimize the blend of hydrocolloids with basis on selected consumer-driven quality properties of bread.

## 2. Materials and Methods

### 2.1. GF Breadmaking Process

GF breads were elaborated with rice flour (Ceifeira, Portugal, 10.4% ± 1.34% moisture, 7.60% ± 0.62% protein, 0.70% ± 0.14% fat and 78.5% carbohydrates, particle size <180 µm, CIELab colorimetric system values of L* = 93.58, a* = 0.12 and b* = 7.43), maize flour (Ceifeira, Portugal, 11.5% ± 1.50% moisture, 5.40% ± 0.52% protein, 2.10% ± 0.28% fat and 78.3% carbohydrates, particle size <180 µm, L*=89.41, a*=0.32 and b*=13.30), white quinoa flour (Incasur, Peru, 8.80% ± 1.03% moisture, 12.9% ± 1.87% protein, 5.00% ± 1.02% fat, 2.20% ashes and 71.1% carbohydrates, particle size <315 µm, L* = 92.5, a* = −1.80 and b* = 27.7) obtained from quinoa grains milled in a laboratory disc mill (Faema, Barcelona, Spain), and then sieved through a rotating sifter. Sunflower oil, white sugar, and refined salt were purchased from a local supermarket (Bragança, Portugal). Instant yeast (*Saccharomyces cerevisiae* lyophilized, Instaferm Lallemand, Setubal, Portugal) and HPMC (E464), XG (E415), and GG (E412) were provided by TecPan (Mirandela, Portugal). All batches were produced using an own recipe with base formulation of rice flour (50%), maize flour (30%) and quinoa flour (20%) to which water (100% flour weight), sunflower oil (6% flour weight), white sugar (3% flour weight), refined salt (1.5% flour weight), instant yeast (3% flour weight) and hydrocolloid mixtures (set at 4% flour weight) were added following a standardized procedure. Demineralized water (pH = 6.8) kept at 5 °C overnight was used. After mixing dry ingredients for 2 min at speed 1, liquid ingredients were added and mixed for an additional 5 min at speed 4, in a professional food processor (SilverCrest SKMP-1200, Bochum, Germany) equipped with a batter blade.

Portions of 250 g were then poured into oiled and floured 520 mL capacity tins, and allowed to proof at 30 °C and 85% of relative humidity for 60 min in a climatic chamber (Climacell 222, MMMGroup, Planegg, Germany). Afterwards, all molds from the same batch were placed in a pre-heated convection oven (2000 W, Princess, Tilburg, The Netherlands) for 60 min at 190 °C. Bread loaves were un-molded after 2 h. All analyses were performed after 24 h.

### 2.2. Analyses of GF Batter and Bread Quality Properties

Batter and crumb rheology parameters were obtained using a texture analyzer TA-XT Plus implemented with the Exponent software version 6.1.11.0 (Stable Micro Systems, Godalming, UK). For calibration, a 5-kg load cell for stickiness analysis, and a 30-kg load cell for back-extrusion and TPA analyses were used

#### 2.2.1. Rheological Properties of GF Batter

For analysis of the batter rheological properties of GF batter, 175 g obtained after the mixing process was used. For the batter stickiness analysis, 5 g batter was weighed in a SMS/Chen-Hoseney stickiness cell (A/DSC) screwed to a SMS/Chen-Hoseney stickiness rig; and then examined by the texture analyzer using a 25-mm perspex cylinder probe (P/25P). Ten repetitions were done for each GF batter formulation. The parameter analyzed was the batter stickiness (STIba, in g) [8].

The back-extrusion analysis was performed using a 35 mm-diameter perspex flat probe (model A/BE), a standard size back-extrusion cylindrical container (50 mm-diameter, capacity of 115 g), and a backward extrusion rig (model A/BE). The container was filled with 85 g of batter (75% full) using a spatula for each analysis to minimize the presence of air bags when filling the container. Four repetitions were tested from each batch of GF batter. The back-extrusion parameter analyzed was the batter firmness (FIRba, in g). The settings of the stickiness and back-extrusion analyses were assessed according to Encina-Zelada et al. [16].

#### 2.2.2. Physicochemical Quality Characteristics of GF Bread

The GF bread loaf volume (mL) was determined using a modified standard rapeseed displacement method 10-05 [17], but using quinoa seeds instead of rapeseeds. The mold used to do the measurement was a parallelepiped with dimensions 10.4 cm × 10.4 cm × 7.5 cm (width × length × height). The loaf specific volume (SVO, mL/g) was calculated as loaf volume divided by loaf weight measured 24 h after baking [18]. Baking loss (BLO) was computed as [(initial loaf weight before baking—the loaf weight after 24 h baking)/initial loaf weight before baking × 100] [19].

Water activity (a_w_) of the bread crumb was measured at 20 °C using an AquaLab (4TE Decagon, MeterGroup, WA, USA), measurements were taken from four central slices from each loaf. Crust color was measured on five different zones of the top of the entire loaf, while crumb color was measured on the center of five slices per each loaf using the L* parameter (luminosity) according to color space CIELab system, with a colorimeter CR-400 (Konica Minolta, chroma meter, Tokyo, Japan) which was calibrated using a white ceramic plate reference (L* = 94.6, a* = −0.46 and b* = 3.88) before each measurement.

#### 2.2.3. Textural Properties of GF Bread Crumb

The characterization of the textural properties of bread crumb was carried out by means of the texture profile analysis (TPA). The GF loaves were sliced using an electric slicer (MAS4000W, Bosch, Stuttgart, Germany) to obtain 20-mm thickness. The two extreme slices of each loaf were discarded, and cylindrical crumb samples of 30 mm-diameter were cut off from the center of each slice. Four sub-samples were analyzed per loaf. For the TPA, a 36 mm-diameter aluminum probe (model P/36R) was used. The settings of the TPA was assessed according to Encina-Zelada et al. [16], with a 50% sample deformation (strain) and double compression (with a gap of 30 s between the two cycles). The TPA parameters analyzed were: hardness (HARbr, in g), adhesiveness (ADHbr, in g·s), cohesiveness (COHbr, dimensionless), and resilience (RESbr, dimensionless). Since ADHbr is calculated from the negative region of the TPA profile curve, the higher the negative values, the more adhesive the crumb.

#### 2.2.4. Image Analysis of GF Bread Crumb Grain Features

Slices of bread were scanned (Pixma MG-2550, Canon, Ho Chi Minh, Vietnam) using the IJ Scan Utility software (version 2.0.12, Canon, Tokyo, Japan) in grey level at −10% brightness, +15% contrast and 350 dpi resolution. Using the ImageJ software (v.1.51j8, Wayne Rasband, National Institute of Health, Bethesda, MD, USA), the center of the bread crumb image was cropped into a 3.9 × 3.9 cm field-of-view (with a spatial resolution of 1 cm = 138 pixels); and saved without any image compression in TIF format for posterior analysis. For each formulation, 24 images (3 loaves × 4 slices × 2 sides of the crumb) were acquired. The grain crumb features of mean cell area (MCA, mm^2^); mean cell density (CDE, cells/mm^2^), and void fraction (VFR, dimensionless), calculated as the proportion of the two-dimensional space occupied by the cells, were computed using the binary segmentation procedure based on the k-means clustering algorithm, proposed in Gonzales-Barron and Butler [20] and Gonzales-Barron and Butler [21], and coded in MATLAB software (ver. R2015a, The Mathworks, Natick, MA, USA).

### 2.3. Experimental Design

#### 2.3.1. Mixture Design

The separate and combined effects of XG, GG, and HPMC gums were assessed using a D-optimal mixture design. Previous breadmaking studies [16,22] aided in the definition of the minimum and maximum doses of XG (i.e., higher doses increased crumb COHbr and RESbr), GG (i.e., higher doses increased crumb COHbr and RESbr), and HPMC (i.e., higher doses increased bread volume) for the current study, in order to avoid bread flaws such as the collapsing of batter matrix during proofing or baking, and the presence of cracked crust, holes in the loaves or excessively dense crumb. The 11 formulations used 100% of water and a total of 4% hydrocolloids consisting of mixtures of HPMC (2.80–3.60%), GG (0.04–0.60%) and XG (0.04–0.60%). The experimental design used in the GF breadmaking in terms of coded and independent variables is shown in Table 1. For each of the 11 treatments, mean and standard deviations of the quality properties of GF batter/bread were calculated; and differences between treatments were assessed by means of one-way analysis of variance (ANOVA), followed by the Tukey’s honest significant difference test (α = 0.05), using Minitab statistical software, version 18.0 free trial (Minitab Inc., Coventry, UK).

#### 2.3.2. Model Fitting

Each of the batter and bread quality parameters were analyzed using a two-factor interaction polynomial linear model, represented by the following general Equation (1)
(1)$$Y_{n}=\beta_{1}\times \mathrm{xanthan}+\beta_{2}\times \mathrm{guar}+\beta_{3}\times \mathrm{HPMC}+\beta_{12}\times \mathrm{xanthan}\times \mathrm{guar}+\beta_{13}\times \mathrm{xanthan}\times \mathrm{HPMC}+\beta_{23}\times \mathrm{guar}\times \mathrm{HPMC}$$
where Y_n_ is the response variable measured for each of the eleven formulations; *β*_1_ is the xanthan regression coefficient; *β*_2_ is the guar regression coefficient; *β*_3_ is the HPMC regression coefficient; *β*_12_ the xanthan × guar interaction coefficient; *β*_13_ the xanthan × HPMC interaction coefficient; and *β*_23_ the guar × HPMC interaction coefficient.

For each quality property, a systematic procedure for model fitting was followed: first, the full model (Equation (1)) was adjusted and the significance (*p* > 0.05) of the terms was appraised. At a second stage, the model was re-fitted without the non-significant terms, in order to improve the precision of the estimates a non-significant linear term remained in the model only if it was significant in interaction with another independent variable. Model lack-of-fit test values were examined by the Fisher’s F-test *p*-value at a level of significance α = 0.05. Normal probability plots of residuals were built to evaluate data normality for all the models. External studentized residuals less than −4 or greater than +4 were considered outliers and were not included in the regression analysis. *R*^2^ and adjusted *R*^2^ were computed to evaluate the fitting quality of potential models; while predicted *R*^2^, adequate precision (AdPrec) and predicted residuals sum of squares (PRESS) were used to evaluate the prediction quality [23,24].

#### 2.3.3. Optimization of Main Dependent Variables of GF Breadmaking

The doses of XG, GG, and HPMC hydrocolloids were optimized based on four quality attributes of bread: specific volume (maximization), crust luminosity (minimization), TPA hardness (minimization), and mean cell density (minimization). In this process, each response variable (Y_n_) is converted into an individual desirability function (d_n_) that varies from 0 to 1; that is, in order to find the factor levels that take to a maximum response variable value, it is necessary to set d_n_ = 1 for high values and d_n_ = 0 for low values of the response variable. Oppositely, for a minimum response variable value, it is necessary to set d_n_ = 0 for high values and d_n_ = 1 for low values [25]. In this way, the desirability function acts as a penalty function that leads the algorithm to regions where the desired response variable values can be found. The factor levels that take to a maximum or a minimum of the response variable are called “optimum points” [26]. Consequently, the design variables are chosen to maximize the overall desirability. The desirability function is expressed as:(2)$$D=\sqrt[{n}]{d_{1}\times d_{2}\times \ldots \times d_{n}}$$
where D is defined as the geometric mean of the individual desirability functions which correspond to variable 1 (d_1_), variable 2 (d_2_), variable “n” (d_n_), and “n” is the number of response variables to optimize. The algorithm should search for response variable values where D tends to 1 [25,26]. Mixture design analysis, optimization, and construction of graphs were carried out using the Design Expert software (version 9.0.6.2, Stat-Ease Inc., Minneapolis, MN, USA).

## 3. Results and Discussion

Means and standard deviation of selected bread quality properties are compiled in Table 2. Analysis of difference of means showed that the softest bread crumb was produced with blends of 3.08–3.48% HPMC, 0.04–0.32% XG, and 0.48–0.60% GG whereas the highest specific volume was obtained with blends of 3.08–3.36% HPMC, 0.60% XG, and 0.04–0.60% GG.

### 3.1. Rheological Properties of GF Batter

The best-fit polynomial models for STIba and FIRba are presented in Table 3. Results show that models of two-factor interaction were adequate to describe all of the GF batter properties. Moreover, according to equation Y_1_, GG in interaction with XG and with HPMC, and XG in interaction with HPMC cause a decreasing effect on STIba—as can be deduced by the negative coefficients—although XG in interaction with GG boosts STIba to a greater extent than GG × HPMC and XG × HPMC interactions. According to equation Y_2_, GG in interaction with XG and with HPMC, and XG in interaction with HPMC increase FIRba; although XG in interaction with GG appears stronger than GG × HPMC and XG × HPMC interactions.

To assess the significance of the GF batter properties’ models, results of ANOVA F-tests are shown in Table 4. For STIba, the model (F-test *p*-value <0.0001) was highly significant for describing the effects of the gums on batter stickiness. Likewise, the *p*-values of the STIba model were all highly significant for the linear mixture terms and the interactions XG × GG, XG × HPMC, and GG × HPMC. The XG × HPMC interaction was a non-significant term which was eliminated from the FIRba model (Table 3).

The model for STIba showed high *R*^2^ (0.8269), *R*^2^ adjusted (0.8185), and *R*^2^ predicted (0.8067) values. On the contrary, low *R*^2^ (0.6456), *R*^2^ adjusted (0.6092), and *R*^2^ predicted values (0.5575) were obtained for FIRba. In relation to the lack-of-fit test for GF batter properties, nearly all (except SCOba model) regression models yielded a significant lack-of-fit (*p* < 0.05), which may indicate that a higher-order model is needed so as to approximate the true response surface [24]. However, because this study attempts to explain the effect of the dependent variables using simple model responses, linear and two-way interaction terms were deemed sufficient. Although not shown here, cubic or quartic models turned the lack-of-fit values non-significant; however, these higher-order models are devoid of real explanation and, hence, were not considered as suitable in this study. The ‘prediction vs. residual’ plots built for each of the final models ailed to show outliers, indicating that models were parsimonious and with acceptable *R*^2^ values.

As regards coefficient of variation (CV), both batter properties analyzed (STIba and FIRba) showed CV values below 10%, having stickiness as the lowest variability (CV = 2.51%). The values of AdPrec, were all higher than 4, indicating that the polynomial models of all GF batter properties could adequately discriminate among treatments. STIba had lower PRESS value than FIRba. For each of the GF batter properties, the model’s F-test was highly significant (*p* < 0.001, in Table 4). The linear mixture terms were significant for STIba. The interaction terms ‘XG × GG’, ‘XG × HPMC’, and ‘GG × HPMC’ were significant (*p* < 0.05) for STIba; while for FIRba, ‘XG × HPMC’ was the only non-significant interaction term (therefore removed from the final model).

Two-dimensional plots (contour plots) and three-dimensional plots are given in Figure 1 for the batter properties. The graphs are shown in five graduated color shades: blue for low values, light blue for intermediate values between low and medium, green for medium values, yellow for values between medium and high, and red for high values [27]. Moreover, correlations between the dependent variable and the independent variables can be read off from the three-D response surface and two-D contour plots. 

The response surfaces suggest that batters of lower stickiness were obtained with a combination of a high XG dose (0.60%) and intermediate doses of HPMC (3.00%) and GG (0.40%), while stickier batters were obtained with a combination of a high GG dose (0.60%), a low XG dose (0.04%) and a high-intermediate HPMC dose (3.36%; Figure 1). For the back-extrusion property analyzed, firmer batters (higher FIRba values) were obtained with a combination of a high-intermediate dose of GG (0.48%) and a low-intermediate dose of HPMC (2.92%) and a high XG dose (0.60%). On the contrary, mixtures consisting of a high GG dose (0.60%), a high-intermediate HPMC dose (3.36%) and a low XG dose (0.04%) produced batters of lower FIRba value.

Although not directly comparable with this study, using a higher level of water (158%) in a mixture of rice/maize/soy (40:40:20), Sciarini et al. [28] obtained lower batter firmness values, ranging from 336 to 679 g, for carrageenan (0.5%) and xanthan (0.5%), respectively. They concluded that the high molecular weight molecules of XG form complex aggregates through hydrogen bonds and polymer entanglements, resulting in a high Newtonian viscosity at low shear rates.

### 3.2. Physicochemical Quality Characteristics of GF Bread

Differences in height and crust appearance of the GF bread treatments can be appreciated in Figure 2. The best-fit polynomial models for SVO, BLO, a_w_, crust L* and crumb L* are presented in Table 3. Results show that models of two-way interactions were adequate to describe all of the GF bread physicochemical properties. GG in interaction with XG and with HPMC causes a synergetic effect on SVO and a_w_; although GG in interaction with HPMC appears stronger than with XG for SVO (equation Y_3_). GG in interaction with HPMC appears slightly stronger than XG × GG interaction for a_w_ (equation Y_5_). In agreement with equation Y_4_, GG interacts with XG and with HPMC to cause a decreasing effect on BLO. GG in interaction with XG boosts BLO more strongly than GG × HPMC interaction. GG in interaction with XG and with HPMC causes a synergetic effect on crust L* (equation Y_6_), and a decreasing effect on crumb L* (equation Y_7_).

The ANOVA F-tests of the GF bread physicochemical models are shown in Table 4. For SVO, the model (F-test *p*-value <0.0001) was highly significant for describing the effects of the gums. Likewise, the *p*-values of the SVO model were significant for the linear mixture terms and the interactions XG × GG and GG × HPMC. Since XG × HPMC was non-significant, it was eliminated from the models for SVO, BLO, a_w_ and crust L*; while the non-significant XG × GG term was removed as a predictor of crumb L*. The GG × HPMC interaction was non-significant for crumb L* (Table 3).

The model for SVO showed very high *R*^2^ (0.9677), *R*^2^ adjusted (0.9666), and *R*^2^ predicted (0.9654) values. On the contrary, low *R*^2^ (0.2161), *R*^2^ adjusted (0.2015), and *R*^2^ predicted (0.1789) values were obtained for crust L*. Among the GF bread physicochemical properties, SVO presented the highest *R*^2^ value (0.9677) compared to BLO (0.7834) and a_w_ (0.5987); while among the colorimetric parameters, crust L* had a moderate *R*^2^ value (0.4944) in comparison with crumb L* (0.2161). In relation to the lack-of-fit test for the GF bread physicochemical properties, nearly all (except BLO model) regression models yielded a highly significant lack-of-fit (*p* < 0.001).

In relation to CV, the lowest CV value was for a_w_ (0.07%), whereas the highest CV value was for baking loss (3.73%); moreover, all GF bread physicochemical properties showed CV values below 10%. Results inferior to 10% might be considered accurate, while values up to 20% are considered acceptable; however, for applications in agriculture, biotechnological processes, microbiology, and clinical protocols, the coefficients of variation are naturally high because of a wide dispersion in data [29]. The values of AdPrec, were all higher than 4, indicating that the polynomial models of all bread GF physicochemical properties could adequately discriminate among treatments. Physicochemical properties of a_w_, SVO and BLO had lower PRESS values than parameters of crust L* and crumb L*.

For each of the GF bread physicochemical properties, the model’s F-test was highly significant (*p* < 0.001, in Table 4). The linear mixture terms were significant (*p* < 0.05) for all physicochemical properties. The interaction term ‘xanthan × guar’ was significant (*p* < 0.05) for SVO, BLO, a_w_ and crust L*; interaction term ‘guar × HPMC’ was significant (*p* < 0.05) for SVO, BLO, a_w_ and crust L*; while interaction term ‘xanthan × HPMC’ was highly significant (*p* < 0.001) for crumb L*. Table 3 displays only the statistically significant coefficients.

The response surfaces modeled suggest that bread loaves of high specific volume can be obtained with a combination of a high XG dose (0.60%) and intermediate doses of HPMC (3.08%) and GG (0.32%; Figure 1). Lower moisture loss during baking were produced with a combination of a high XG dose (0.60%) and intermediate doses of GG (0.37%) and HPMC (3.03%). Lower crumb a_w_ values were obtained with a combination of a low HPMC dose (2.80%) and high doses of XG (0.60%) and GG (0.60%). Regarding the colorimetric properties of GF bread crust, lower crust L* values (i.e., darker crust) were produced with a combination of a low XG dose (0.04%), a high GG dose (0.60%) and a high-intermediate HPMC dose (3.36%). In relation to the crumb color, higher crumb L* values (i.e., lighter crumb) were obtained with a combination of a high XG dose (0.60%), and intermediate doses of GG (0.26%) and HPMC (3.14%).

Evaluating GF breads from different specialized supermarket sources, Matos and Rosell [30] measured higher values of SVO (1.54–4.77 mL/g) and a_w_ (0.89–0.97 g) than those found in the present study; and found values of crumb L* (65.8–80.2) that were comparable to those measured in this study. Pongjaruvat et al. [31] formulated GF breads varying pregelatinized tapioca starch doses, and obtained lower values for SVO (1.80–2.70 mL/g), similar values for crust L* (55.7–70), crumb L* (62.9–70.8), in a recipe of jasmine rice flour (100%), water content (80%), shortening (10%), sugar (9%), soy flour (10%), yeast (1.5%), and salt (1.5%). Bourekoua et al. [3] studied different gums and water content (WC) in GF breadmaking formulations based on a mixture rice/field bean flour using a screening design, and obtained an optimum formula with a combination of 1.5% of gum arabic and 71.5% WC, obtaining a similar value for SVO (2.87 mL/g) and a darker crumb with a lower L* value (60.3). Demirkesen et al. [4] studied different gums and emulsifiers in the elaboration of GF breads based on 100% rice flour and 150% water content, obtaining lower SVO values from 0.6 to 1.6 mL/g.

In this specific topic, considering that from a consumer, producer, and food safety points of view, larger-sized breads with lower baking losses, lower water activity crumbs, and darker crusts are preferred to smaller breads with higher a_w_ and lighter crusts. Gum blends of low HPMC dose (2.80%) and high doses of XG (0.60%) and GG (0.60%) would appear as adequate since they lead to GF breads of higher SVO (2.85 mL/g) and lower values of BLO (12.6%), a_w_ (0.982) and crust L* (68.7).

### 3.3. Textural Properties of GF Bread Crumb

According to Table 5, some TPA properties were more discriminant than others: SPRbr, COHbr, and RESbr appeared as the most discriminant parameters capable of statistically distinguishing the different blends (Table 5). Two-way interaction models were appropriate to describe the GF bread crumb textural properties of HARbr, ADHbr, COHbr, and RESbr (Table 6). Moreover, in agreement with equations Y_8_ and Y_9_, GG in interaction with both XG and with HPMC causes a synergetic effect on HARbr and ADHbr, respectively; however, GG × XG has a stronger effect on HARbr than GGxHPMC, and the opposite is true for ADHbr. According to equation Y_10_, GG in interaction with XG and with HPMC, and XG in interaction with HPMC cause a decreasing effect on COHbr; although XG in interaction with HPMC increases COHbr to a greater extent than XG × GG and GG × HPMC interactions, respectively. GG in interaction with XG has a stronger effect than in interaction with HPMC (equation Y_10_). In relation to equation Y_11_, GG in interaction with XG and with HPMC, and XG in interaction with HPMC causes a decreasing effect on RESbr; although XG in interaction with GG improves RESbr more strongly than both XG × HPMC and GG × HPMC interactions.

The ANOVA F-tests of textural properties of GF bread crumb models are shown in Table 7. For HARbr, the model (F-test *p*-value <0.0001) was highly significant for describing the individual effect (linear mixture terms) and the combined effects of interactions: XG × GG and GG × HPMC. In the prediction of HARbr, ADHbr, and RESbr, the XG × HPMC interaction was not significant. Likewise, the GG × HPMC interaction was eliminated from the RESbr model (Table 6). The model for ADHbr showed high values of *R*^2^ (0.7741), *R*^2^ adjusted (0.7670) and *R*^2^ predicted (0.7575). On the contrary, the lowest *R*^2^ (0.2076), *R*^2^ adjusted (0.189), and *R*^2^ predicted (0.1513) values were obtained for RESbr. Among the GF bread crumb textural properties, ADHbr presented the highest *R*^2^ value (0.7741) compared to HARbr (0.6602), COHbr (0.2849), and RESbr (0.2076). In relation to the lack-of-fit test for GF bread crumb textural properties, all regression models yielded a highly significant lack-of-fit (*p* < 0.001), most likely due to data variability and the non-use of higher order terms, as explained before.

As regards CV, COHbr presented the lowest value (3.29%), while the highest one was for ADHbr (42.1%). Furthermore, COHbr and RESbr (6.49%) showed CV values below 10%, whereas HARbr (16.8%) and ADHbr presented CV values higher than 10%. The values of AdPrec, were all higher than 4, indicating that the polynomial models of all GF bread crumb textural properties could adequately discriminate among treatments. Crumb textural properties of COHbr and RESbr had lower PRESS values than HARbr and ADHbr. For each of the GF bread crumb textural properties, the model’s F-test and the linear mixture terms were highly significant (*p* < 0.0001, in Table 7). The interaction term ‘XG × HPMC’ was highly significant (*p* < 0.0001) only for COHbr; and the interaction term ‘GG × HPMC’ was significant (*p* < 0.05) for HARbr, ADHbr, and COHbr. 

The response surfaces suggest that softer crumbs were obtained with a combination of a high-intermediate HPMC dose (3.36%), a high GG dose (0.60%) and a low XG dose (0.04%; Figure 3). Mixtures consisting of a low XG dose (0.04%), a high-intermediate HPMC dose (3.36%) and a high dose of GG (0.60%) rendered lower ADHbr values. Higher crumb cohesiveness values can be obtained with combinations of high doses of XG (0.60%) and GG (0.60%) and a low HPMC dose (2.80%). Higher crumb resilience values were obtained with combinations of a high XG dose (0.60%), a low GG dose (0.04%), and a high-intermediate HPMC dose (3.36%).

According to Cornejo and Rosell [32], high resilience values are preferred because it is a property related to the bread freshness and elasticity, as perceived by the consumers. A reduction in resilience characterizes loss of elasticity since that property indicates the ability of a material to return to its original shape after stressing. On the other hand, bread crumb with high cohesiveness is desirable because it forms a bolus, instead of disintegrating during mastication whereas low cohesiveness indicates increased susceptibility of the bread to fracture or crumble. Morreale et al. [33] obtained, from GF bread elaborated with rice flour, lower values of hardness (145–267 g), comparable value of cohesiveness (0.748–0.805) and higher values of resilience (0.361–0.386). From GF bread elaborated with a base of rice flour and potato starch, Turkut et al. (2016) obtained a comparable crumb hardness (1672–3569 g) and cohesiveness (0.42–0.47), when they varied the levels of quinoa and buckwheat flours. In another study, Hager et al. [34] obtained similar TPA hardness values (1916–6797 g). Bourekoua et al. [3] obtained softer crumbs (1523 g); likewise, Demirkesen et al. [4] obtained lower firmness (i.e., hardness) values from 25 to 255 g.

In this specific topic, since from a consumer standpoint, softer and more cohesive and elastic crumbs are preferred than harder and stiffer crumbs, combinations of a high HPMC dose (3.60%), a low XG dose (0.04%), and an intermediate GG dose (0.36%) would be appropriate to obtain GF bread crumbs of higher values of COHbr (0.433) and RESbr (0.167) and lower values of HARbr (2362 g) and ADHbr (−163 g·s).

### 3.4. Image Analysis of GF Bread Crumb Grain Features

Differences in crumb grain features of the GF bread treatments can be appreciated in Figure 4. All of the crumb grain features were appropriately described by the response surface models (Table 6). Furthermore, according to equation Y_12_, GG in interaction with XG causes a decreasing effect on MCA. GG in interaction with XG and with HPMC causes a synergetic effect on CDE (equation Y_13_), while XG in interaction with HPMC causes an opposite effect on CDE. GG interacts with XG to affect CDE to a greater extent than GG × HPMC and XG×HPMC do. According to equation Y_14_, GG in interaction with XG and with HPMC causes a decreasing effect on VFR; although GG in interaction with XG has a stronger effect than in interaction with HPMC.

The ANOVA F-tests of the GF bread crumb grain features’ models are shown in Table 7. For crumb cell density, the model (F-test *p*-value <0.0001) was highly significant for describing the individual effect (linear mixture terms) and the combined effects of interactions of XG × GG, XG × HPMC and GG × HPMC. Since XG × HPMC interaction was non-significant, it was eliminated from the VFR model (Table 6). The model for VFR showed high values of *R*^2^ (0.6919), *R*^2^ adjusted (0.6871), and *R*^2^ predicted (0.6812). On the contrary, the lowest *R*^2^ (0.6513), *R*^2^ adjusted (0.6445), and *R*^2^ predicted (0.6337) values were obtained for CDE. Among the GF bread crumb grain features, VFR presented the highest *R*^2^ value (0.6919) in comparison with MCA (0.6522) and CDE (0.6513). In relation to the lack-of-fit test for the GF bread crumb grain features, all regression models yielded a highly significant lack-of-fit (*p* < 0.0001).

In relation to CV, the lowest CV value was for VFR (8.92%), while the highest one was for MCA (31.2%). Moreover, only VFR (8.92%) showed a CV value below 10%, whereas CDE (21.1%) and MCA had values higher than 10% of CV. Crumb grain features of CDE, VFR, COM, and ARA had lower PRESS values than MCA and UNI. For each of the GF bread crumb grain features, the model’s F-test and the linear mixture terms were highly significant (*p* < 0.0001, in Table 7). All the crumb grain features were affected by the interaction XG × GG (*p* < 0.05). The interaction term XG × HPMC was significant (*p* < 0.05) for CDE, UNI, COM, and ARA; while the interaction term GG × HPMC was significant (*p* < 0.05) for CDE, UNI, and VFR.

The response surfaces suggest that crumb grain of higher mean cell area (i.e., more open crumb grains) can be obtained with a combination of a low XG dose (0.04%), a high GG dose (0.60%), and a high-intermediate HPMC dose (3.36%; Figure 5). Higher cell density values (i.e., denser crumb appearance) were produced with combinations of a high XG dose (0.60%) and intermediate doses of GG (0.36%) and HPMC (3.04%). Higher crumb void fraction values (i.e., greater porosity structure) were obtained with a combination of a high GG dose (0.60%), a high-intermediate HPMC dose (3.36%) and a low XG dose (0.04%).

Bourekoua et al. [3] obtained a denser crumb appearance with a lower value of MCA (1.87 mm^2^). Sciarini et al. [35], formulating GF breads with hydrocolloids from different sources, obtained lower values for crumb UNI (ranged from 1.42 to 2.07), comparable values for MCA (2.73–8.06 mm^2^), and higher values for CDE (0.89–1.67 cells/mm^2^), in a matrix of rice flour (45%), cassava starch (45%), soy flour (10%), WC (75%), and hydrocolloid (0.5%). As found in this study, they also concluded that GF breads often produce a dense crumb structure with thick cell walls; hence, the presence of larger cells producing a spongier crumb is not easily obtained in GF breads.

In this specific topic, considering that from a consumer viewpoint, bread loaves of more open crumbs and greater cell size (Figure 4; formulations 3, 8, or 9) are preferred than compact, denser and closer crumbs of smaller pores (Figure 4; formulations 5, 10, or 11), blends of a low XG dose (0.04%), a high GG dose (0.60%), and a high-intermediate HPMC dose (3.36%) would be suitable to achieve GF bread crumbs of lower value of CDE (0.10 cells/ mm^2^), and higher values of MCA (4.17 mm^2^) and VFR (0.45).

### 3.5. Optimization of the Gum Mixtures on Main GF Bread Quality Parameters

The quality requirements for bread production can be set from the producer and/or the consumer perspective. One of the most important parameters in GF breadmaking, known to strongly influence consumer’s choice, is the loaf specific volume, as larger loaves are perceived as more appealing. According to Dapčević-Hadnađev et al. [36], the quality characteristics that determine the choice of bread by the consumer are those related to its appearance, such as the size (volume) of the bread, color (crust color), bread shape, and freshness (related with the crumb softness), which frequently is checked by pressing the bread. As stated by Živančev, et al. [37], the crust color of bread is a very important parameter for consumers because the acceptability of the product strongly depends on a golden brown color [38]. Furthermore, bread crumb grain, perceived by the consumers as degree of porosity, is another important quality characteristic which indicates softness and foamy crumb structure [38]. Therefore, with basis on the information above, to determine the optimal hydrocolloids mixture, the optimization was carried out by maximizing the bread specific volume (mL/g), and at the same time minimizing the crust luminosity, crumb hardness (g) and mean cell density (cells/mm^2^). The bread quality features to optimize were assigned the same relative importance. The optimal formulation of 3.16% HPMC, 0.60% GG, and 0.24% XG produced a desirability of 0.54. This optimized mixture of gums would produce in the final bread product, the following estimated values for specific volume: 2.57 mL/g; crust L*: 67.0; crumb hardness: 2082 g; and mean cell density: 0.142 cells/mm^2^. The maximum prediction point indicated in Figure 6 was determined by numerical optimization of the polynomial models, by setting the factors goals ‘within the range’ and the variable responses goal to ‘maximize’ or ‘minimize’.

In relation to the hydrocolloids’ interaction effect on the GF bread properties, it was found that XG × GG interacted to increase batter firmness, bread specific volume, baking loss, crumb a_w_, crust lightness, crumb hardness, adhesiveness, mean cell density, and cell size uniformity; while decreasing batter stickiness, crumb cohesiveness, resilience, mean cell area, and void fraction. XG interacted with HPMC to increase batter firmness, bread crumb hardness, and adhesiveness; and to reduce batter stickiness, cohesiveness, resilience, and mean cell density, cell size uniformity, and void fraction. Furthermore, GG also interacted with HPMC to increase batter firmness, bread specific volume, crumb a_w_, crust and crumb lightness, crumb hardness, adhesiveness, mean cell density, and cell size uniformity; and to decrease batter stickiness, bread baking loss, cohesiveness, and resilience. The three-way interaction XG × GG × HPMC was not assessed in this study.

## 4. Conclusions

It can be concluded that the quality of the final gluten-free bread is affected by the interactions that take place between the three hydrocolloids. Xanthan in interaction with guar, and guar in interaction with HPMC improved the bread specific volume, the softness and the mean cell density, associated with more open crumb grain structure. On the other hand, xanthan in interaction with HPMC produced batter of decreased stickiness, crumb cohesiveness and resilience, and mean cell density, associated with denser crumb appearance. Sticky batters were produced with a combination of high dose of GG (0.60%), high-intermediate dose of HPMC (3.36%), and low dose of XG (0.04%). GF breads of greater specific volume were obtained with a combination of high XG dose (0.60%) and intermediate doses of HPMC (3.08%) and GG (0.32%). Darker bread crust was obtained with a mixture low XG dose (0.04%), high-intermediate HPMC dose (3.36%) and high GG dose (0.60%); while a mixture of high-intermediate HPMC dose (3.36%), high GG dose (0.60%), and low XG dose (0.04%) rendered desirable softer crumbs and bread slices of more open visual texture. From a desirability approach, the optimum hydrocolloid mixture was obtained with a blend of 0.24% XG, 0.60% GG, and 3.16% HPMC, which would yield loaves of good quality in terms of high specific volume (2.57 mL/g), and low values of crust luminosity (L* = 67.0), crumb hardness (2082 g), and mean cell density (0.14 cells/mm^2^). Further studies can be conducted to validate the optimal formulation.

## Figures and Tables

**Figure 1 foods-08-00156-f001:**
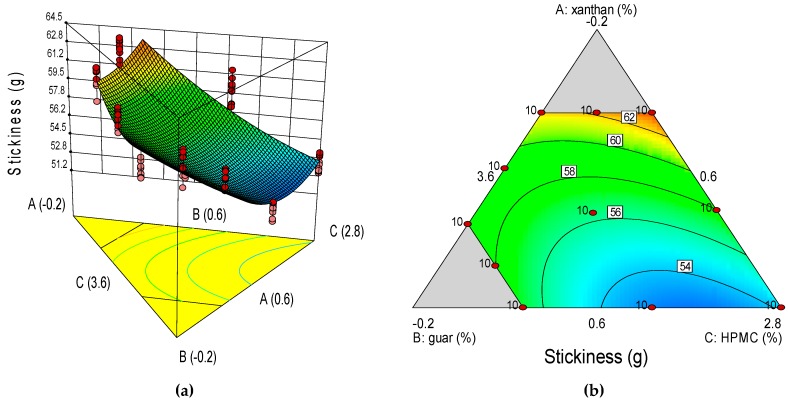
Three-dimensional surface (3-D, left) and two-dimensional contour (2-D, right) plots showing the effect of xanthan gum (A), guar gum (B), and HPMC (C) on GF batter and bread physicochemical properties, were: (**a**), stickiness 3-D; (**b**), stickiness 2-D; (**c**), firmness 3-D; (**d**), firmness 2-D; (**e**), specific volume 3-D; (**f**), specific volume 2-D; (**g**), baking loss 3-D; (**h**), baking loss 2-D; (**i**), a_w_ 3-D; (**j**), a_w_ 2-D; (**k**), crumb L* 3-D; (**l**), crumb L* 2-D; (**m**), crust L* 3-D; and (**n**), crust L* 2-D.

**Figure 2 foods-08-00156-f002:**
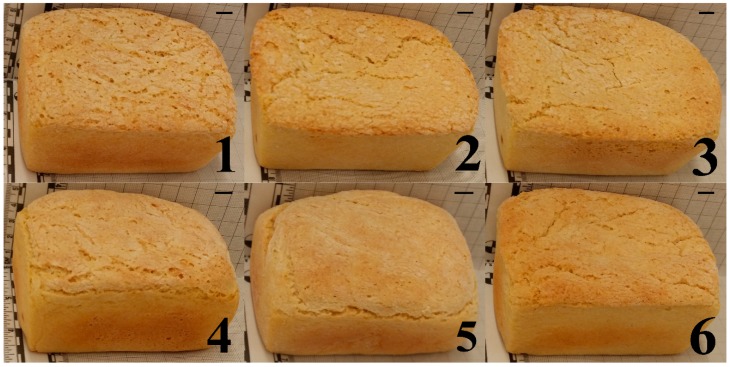
Height and crust appearance of the eleven treatments of gluten-free bread produced by varying xanthan, guar and HPMC hydrocolloids. Formulations correspond to those shown in Table 1 (Scale bars = 1 cm).

**Figure 3 foods-08-00156-f003:**
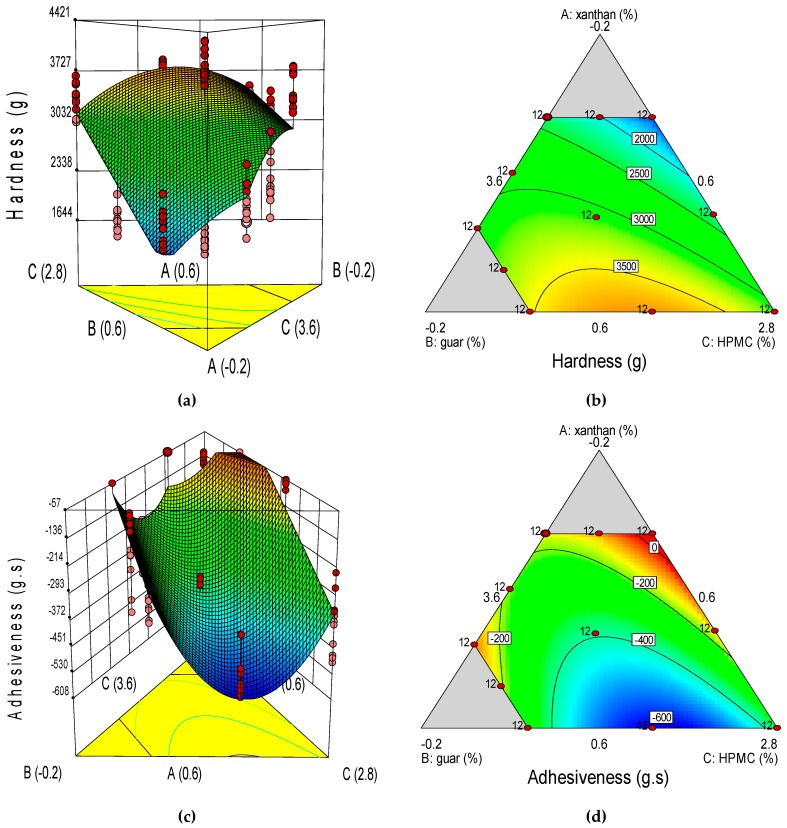
Three-dimensional surface (3-D, left) and two-dimensional contour (2-D, right) plots showing the effect of xanthan (A), guar (B), and HPMC (C) gums on bread textural profile analysis (TPA) of hardness (g), adhesiveness (g·s), cohesiveness (dimensionless) and resilience (dimensionless), were: (**a**), hardness 3-D; (**b**), hardness 2-D; (**c**), adhesiveness 3-D; (**d**), adhesiveness 2-D; (**e**), cohesiveness 3-D; (**f**), cohesiveness 2-D; (**g**), resilience 3-D; and (**h**), resilience 2-D.

**Figure 4 foods-08-00156-f004:**
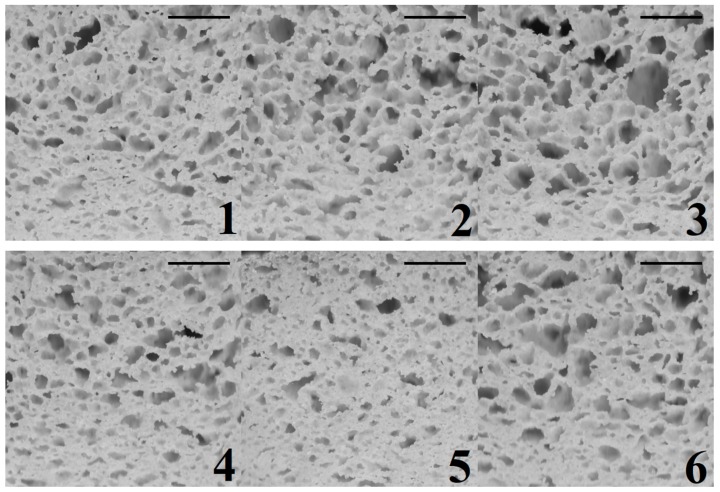
Crumb grain images of the 11 treatments of gluten-free bread produced by varying xanthan, guar, and HPMC hydrocolloids. Numbers correspond to the formulations shown in Table 1 (Scale bars = 1 cm).

**Figure 5 foods-08-00156-f005:**
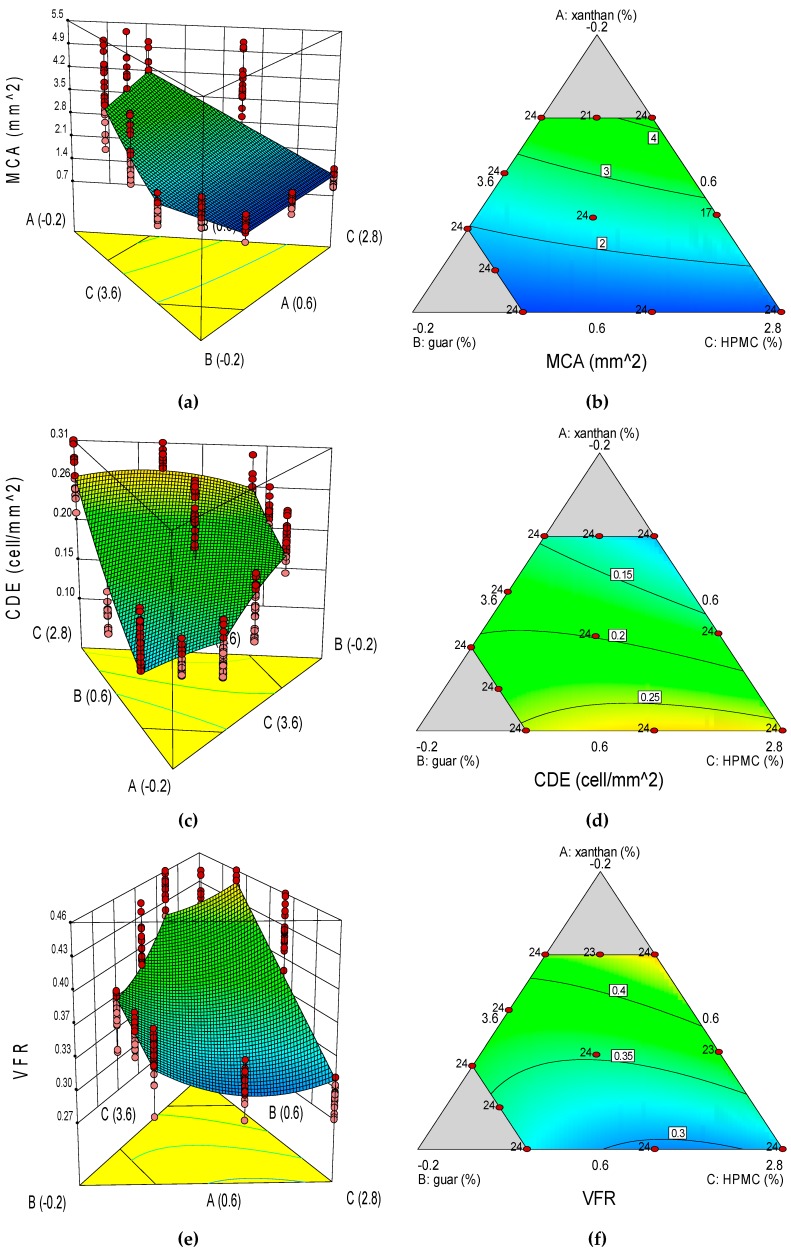
Three-dimensional surface (3-D, left) and two-dimensional contour (2-D, right) plots showing the effect of xanthan gum (A), guar gum (B) and HPMC (C) on the gluten-free crumb grain features of mean cell area (MCA, mm^2^); mean cell density (CDE, cells/mm^2^); void fraction (VFR, dimensionless), were: (**a**), MCA 3-D; (**b**), MCA 2-D; (**c**), CDE 3-D; (**d**), CDE 2-D; (**e**), VFR 3-D; and (**f**), VFR 2-D.

**Figure 6 foods-08-00156-f006:**
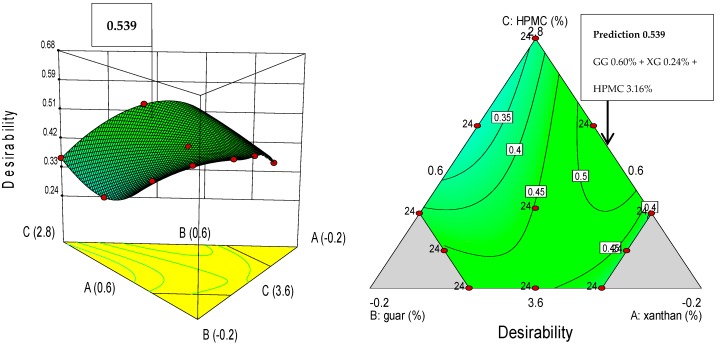
Three-dimensional desirability (**left**) and two-dimensional desirability contour (**right**) plots of the predicted optimal solution of xanthan, guar, and HPMC gums determined by desirability function analysis.

**Table 1 foods-08-00156-t001:** Experimental design showing the doses of the three hydrocolloids (xanthan: XG, guar: GG and HPMC) used in the elaboration of GF bread.

Mixture Formulation	Fraction (Coded) Values			Independent Variables		
	XG	GG	HPMC	Xanthan (%)	Guar (%)	HPMC (%)
1	0.09	0.01	0.90	0.36	0.04	3.60
2	0.01	0.09	0.90	0.04	0.36	3.60
3	0.01	0.15	0.84	0.04	0.60	3.36
4	0.15	0.01	0.84	0.60	0.04	3.36
5	0.15	0.15	0.70	0.60	0.60	2.80
6	0.05	0.05	0.90	0.20	0.20	3.60
7	0.12	0.01	0.87	0.48	0.04	3.48
8	0.08	0.15	0.77	0.32	0.60	3.08
9	0.01	0.12	0.87	0.04	0.48	3.48
10	0.15	0.08	0.77	0.60	0.32	3.08
11	0.08	0.08	0.84	0.33	0.33	3.34

**Table 2 foods-08-00156-t002:** Mean values and standard deviations (in brackets) of the gluten-free batter rheological properties of stickiness and firmness and physicochemical properties of gluten-free bread by formulation

Formulation	Doses of Hydrocolloids (%)			Rheological Properties of Batter and Physicochemical Properties of Bread						
	Xanthan	Guar	HPMC	Stickiness (g)	Firmness (g)	Specific Volume (mL/g)	Baking Loss (%)	a_w_	Crust L*	Crumb L*
1	0.36	0.04	3.60	57.8 ^e^ (0.591)	2508 ^cd^ (50.7)	2.69 ^d^ (0.09)	13.6 ^bcd^ (0.135)	0.9843 ^ab^ (0.0004)	67.2 ^c^ (0.662)	70.4 ^de^ (1.981)
2	0.04	0.36	3.60	62.0 ^b^ (0.921)	2699 ^bc^ (57.6)	2.43 ^i^ (0.05)	14.6 ^ab^ (0.393)	0.9848 ^a^ (0.0003)	65.8 ^de^ (0.861)	71.4 ^cde^ (2.27)
3	0.04	0.60	3.36	61.5 ^b^ (0.662)	2202 ^e^ (38.5)	2.41 ^i^ (0.01)	15.0 ^a^ (0.310)	0.9832 ^c^ (0.0003)	66.4 ^cde^ (1.151)	72.4 ^bcd^ (1.352)
4	0.60	0.04	3.36	57.8 ^e^ (0.613)	2259 ^e^ (110)	2.85 ^b^ (0.08)	13.1 ^cde^ (0.321)	0.9839 ^bc^ (0.0006)	67.2 ^c^ (0.481)	71.2 ^cde^ (1.172)
5	0.60	0.60	2.80	53.9 ^f^ (0.660)	2905 ^ab^ (165)	2.83 ^c^ (0.07)	12.7 ^de^ (0.173)	0.9813 ^c^ (0.0004)	69.8 ^a^ (0.762)	72.0 ^bcd^ (1.861)
6	0.20	0.20	3.60	60.3 ^c^ (0.711)	2595 ^c^ (75.7)	2.58 ^g^ (0.06)	12.9 ^cde^ (0.204)	0.9848 ^a^ (0.0005)	66.2 ^cde^ (1.201)	69.5 ^e^ (2.371)
7	0.48	0.04	3.48	58.7 ^de^ (1.013)	2346 ^de^ (80.6)	2.70 ^d^ (0.01)	13.6 ^bcd^ (0.211)	0.9838 ^bc^ (0.0011)	66.6 ^cd^ (0.921)	73.7 ^ab^ (1.112)
8	0.32	0.60	3.08	59.7 ^cd^ (1.032)	2170 ^e^ (45.1)	2.65 ^e^ (0.03)	14.1 ^abc^ (1.139)	0.9832 ^c^ (0.0004)	66.5 ^cde^ (0.992)	70.8 ^cde^ (2.623)
9	0.04	0.48	3.48	63.7 ^a^ (0.781)	2177 ^e^ (111)	2.47 ^h^ (0.09)	14.9 ^a^ (0.588)	0.9833 ^c^ (0.0004)	65.5 ^e^ (1.372)	72.9 ^abc^ (1.971)
10	0.60	0.32	3.08	52.4 ^g^ (0.571)	2623 ^c^ (37.8)	2.92 ^a^ (0.01)	11.9 ^e^ (0.239)	0.9835 ^c^ (0.0010)	68.5 ^b^ (0.561)	74.8 ^a^ (1.472)
11	0.33	0.33	3.34	55.0 ^f^ (0.472)	2988 ^a^ (116)	2.63 ^f^ (0.08)	13.3 ^cd^ (0.258)	0.9834 ^c^ (0.0004)	69.0 ^ab^ (0.731)	73.9 ^ab^ (1.361)

Different superscripts letters (^a,b,c,d,e,f,g,h,i^) within a column indicate significant differences (*p* ≤ 0.05) between formulations.

**Table 3 foods-08-00156-t003:** Predicted polynomial models showing the effect of each mixture component and their significant interactions on gluten-free batter rheological properties of stickiness and firmness and gluten-free bread physicochemical properties

Parameters	Predicted Model Equations
Stickiness (g)	Y_1_ = 43.1xanthan + 121.7guar + 16.7HPMC − 48.3xanthan × guar − 12.1xanthan × HPMC − 32.0guar × HPMC
Firmness (g)	Y_2_ = −502xanthan − 11241guar + 713HPMC + 7354xanthan × guar + 3087guar × HPMC
Specific volume (mL/g)	Y_3_ = 1.38xanthan − 1.03guar + 0.59HPMC + 0.44xanthan × guar + 0.49guar × HPMC
Baking loss (%)	Y_4_ = 1.92xanthan + 37.1guar + 3.59HPMC − 14.8xanthan × guar − 9.2guar × HPMC
a_w_	Y_5_ = 0.2439xanthan + 0.2252guar + 0.2463HPMC + 0.0048xanthan × guar + 0.0053guar × HPMC
Crust L*	Y_6_ = 18.9xanthan − 4.61guar + 16.4HPMC + 13.1xanthan × guar + 6.12guar × HPMC
Crumb L*	Y_7_ = 29.9xanthan − 56.1guar + 16.1HPMC + 25.1guar × HPMC

**Table 4 foods-08-00156-t004:** Summary statistics of the analysis of variance (ANOVA) showing F-test and *p*-value (in brackets, Prob > F) of the full model, linear mixture terms, interaction terms and lack-of-fit from polynomial models fitted on the GF batter rheological properties of stickiness and firmness and GF bread physicochemical properties. Goodness-of-fit measures including coefficient of variation (CV), predicted residual error sum of squares (PRESS), coefficient of determination (*R*^2^), adjusted *R*^2^ (*R*^2^ adj), predicted *R*^2^ (*R*^2^ pred), and adequate precision (AdPrec) are also shown.

Dependent Variables	ANOVA						Fitting Quality					
	Model	Linear Mixture Terms	Xanthan × Guar	Xanthan × HPMC	Guar × HPMC	Lack-of-Fit	CV (%)	PRESS	*R* ^2^	*R*^2^ adj	*R*^2^ pred	AdPrec
Stickiness	99.3 (<0.0001)	211 (<0.0001)	61.7 (<0.0001)	6.66 (0.01)	46.7 (<0.0001)	60.3 (<0.0001)	2.51	251	0.8269	0.8185	0.8067	30.7
Firmness	17.8 (<0.0001)	3.60 (0.036)	61.5 (<0.0001)	-	12.3 (0.001)	21.5 (<0.0001)	7.30	1.62×10^6^	0.6456	0.6092	0.5575	12.3
Specific volume	950 (<0.0001)	1881 (<0.0001)	24.7 (0.0002)	-	35.3 (<0.0001)	113 (<0.0001)	1.12	0.12	0.9677	0.9666	0.9654	79.5
Baking loss	25.3 (<0.0001)	38.5 (<0.0001)	24.2 (<0.0001)	-	10.7 (0.003)	2.20 (0.08)	3.73	9.51	0.7834	0.7525	0.7145	16.7
a_w_	47.4 (<0.0001)	90.5 (<0.0001)	5.64 (0.02)	-	7.70 (0.006)	9.52 (<0.0001)	0.07	0.00007	0.5987	0.5861	0.5678	20.7
Crust L*	39.1 (<0.0001)	68.7 (<0.0001)	18.7 (<0.0001)	-	4.65 (0.03)	15.3 (<0.0001)	1.70	219	0.4944	0.4818	0.4662	19.1
Crumb L*	14.8 (<0.0001)	4.39 (0.01)	-	35.6 (<0.0001)	-	8.45 (<0.0001)	2.94	758	0.2161	0.2015	0.1789	12.9

-, Term dropped from equation due to non-significance (*p* > 0.05).

**Table 5 foods-08-00156-t005:** Mean values and standard deviations (in brackets) of the gluten-free bread crumb textural properties and crumb grain image analysis features extrusion by formulation

Formulation	Doses of Hydrocolloids (%)			Texture Profile Analysis and Image Analysis Features of Bread Crumb						
	Xanthan	Guar	HPMC	Hardness (g)	Adhesiveness (g·s)	Cohesiveness	Resilience	Mean Cell Area (mm^2^)	Cell Density (cells/mm^2^)	Void Fraction
1	0.36	0.04	3.60	3511 ^bc^ (227)	−40 ^a^ (14.5)	0.432 ^bc^ (0.008)	0.171 ^ab^ (0.005)	1.73 ^e^ (0.23)	0.228 ^cd^ (0.023)	0.350 ^d^ (0.015)
2	0.04	0.36	3.60	2224 ^ef^ (266)	−34 ^a^ (9.23)	0.422 ^c^ (0.024)	0.170 ^ab^ (0.005)	4.10 ^bc^ (0.88)	0.125 ^fg^ (0.022)	0.446 ^a^ (0.029)
3	0.04	0.60	3.36	2026 ^fg^ (196)	−22 ^a^ (7.98)	0.430 ^bc^ (0.023)	0.175 ^a^ (0.006)	3.49 ^cd^ (0.77)	0.140 ^ef^ (0.023)	0.427 ^b^ (0.025)
4	0.60	0.04	3.36	3179 ^d^ (197)	−291 ^c^ (57.5)	0.454 ^a^ (0.009)	0.177 ^a^ (0.006)	1.52 ^e^ (0.18)	0.245 ^bc^ (0.018)	0.333 ^d^ (0.017)
5	0.60	0.60	2.80	3288 ^cd^ (225)	−389 ^d^ (86.1)	0.453 ^a^ (0.009)	0.172 ^ab^ (0.005)	1.14 ^e^ (0.17)	0.285 ^a^ (0.037)	0.288 ^f^ (0.011)
6	0.20	0.20	3.60	2425 ^e^ (301)	−313 ^cd^ (55.9)	0.445 ^ab^ (0.012)	0.173 ^a^ (0.005)	2.81 ^d^ (0.67)	0.162 ^e^ (0.028)	0.394 ^c^ (0.020)
7	0.48	0.04	3.48	3189 ^d^ (162)	−204 ^b^ (98.2)	0.438 ^abc^ (0.014)	0.167 ^ab^ (0.006)	1.76 ^e^ (0.26)	0.218 ^d^ (0.027)	0.341 ^d^ (0.012)
8	0.32	0.60	3.08	1732 ^gh^ (267)	−43 ^a^ (23.6)	0.425 ^c^ (0.015)	0.155 ^cd^ (0.020)	5.00 ^e^ (2.04)	0.108 ^g^ (0.026)	0.441 ^ab^ (0.041)
9	0.04	0.48	3.48	1717 ^h^ (202)	−91 ^a^ (38.1)	0.434 ^bc^ (0.021)	0.147 ^d^ (0.013)	4.57 ^ab^ (1.34)	0.109 ^g^ (0.022)	0.426 ^ab^ (0.028)
10	0.60	0.32	3.08	3733 ^ab^ (95)	−558 ^e^ (52.5)	0.440 ^abc^ (0.007)	0.166 ^abc^ (0.004)	1.30 ^e^ (0.17)	0.263 ^ab^ (0.029)	0.305 ^ef^ (0.013)
11	0.33	0.33	3.34	3868 ^a^ (206.2)	−496 ^e^ (90.10)	0.426 ^c^ (0.012)	0.161 ^bc^ (0.004)	1.36 ^e^ (0.18)	0.256 ^b^ (0.029)	0.310 ^e^ (0.011)

Different superscripts letters (^a,b,c,d,e,f,g,h^) within a column indicate significant differences (*p* ≤ 0.05) between formulations.

**Table 6 foods-08-00156-t006:** Predicted polynomial models showing the effect of each mixture component and their significant interactions on gluten-free bread crumb textural properties and crumb grain image analysis features

Parameters	Predicted Model Equations
Hardness (g)	Y_8_ = 2311xanthan − 20773guar + 580HPMC + 7799xanthan × guar + 5825guar × HPMC
Adhesiveness (g·s)	Y_9_ = −795xanthan + 12163guar + 81HPMC − 3439xanthan × guar – 3665guar × HPMC
Cohesiveness	Y_10_ = 0.658xanthan + 0.348guar + 0.116HPMC − 0.154xanthan × guar − 0.166xanthan × HPMC − 0.085guar × HPMC
Resilience	Y_11_ = 0.395xanthan + 0.334guar + 0.049HPMC − 0.137xanthan × guar − 0.112xanthan × HPMC − 0.099guar × HPMC
Mean cell area (mm^2^)	Y_12_ = −2.035xanthan + 3.154guar + 0.73HPMC − 3.866xanthan × guar
Cell density (cells/mm^2^)	Y_13_ = 0.83xanthan − 0.758guar + 0.042HPMC + 0.259xanthan × guar − 0.192xanthan × HPMC + 0.197guar × HPMC
Void fraction	Y_14_ = −0.064xanthan + 1.089guar + 0.108HPMC − 0.412xanthan × guar − 0.274guar × HPMC

**Table 7 foods-08-00156-t007:** Summary statistics of the analysis of variance (ANOVA) showing F-test and *p*-value (in brackets, Prob > F) of the full model, linear mixture terms, interaction terms and lack-of-fit from polynomial models fitted on GF bread crumb textural properties and crumb grain features image analysis. Goodness-of-fit measures including coefficient of variation (CV), predicted residual error sum of squares (PRESS), coefficient of determination (*R*^2^), adjusted *R*^2^ (*R*^2^ adj), predicted *R*^2^ (*R*^2^ pred), and adequate precision (AdPrec) are also shown.

Dependent Variables	ANOVA					Fitting Quality						
	Model	Linear Mixture Terms	Xanthan × Guar	Xanthan × HPMC	Guar × HPMC	Lack-of-Fit	CV (%)	PRESS	R^2^	R^2^ adj	R^2^ pred	AdPrec
Hardness	61.7 (<0.0001)	107 (<0.0001)	31.1 (<0.0001)	-	19.7 (<0.0001)	77.3 (<0.0001)	16.8	3.02×10^7^	0.6602	0.6495	0.6364	24.6
Adhesiveness	109 (<0.0001)	109 (<0.0001)	149 (<0.0001)	-	193 (<0.0001)	36.9 (<0.0001)	42.1	1.23×10^5^	0.7741	0.7670	0.7575	36.5
Cohesiveness	10.0 (<0.0001)	16.7 (<0.0001)	7.86 (0.006)	15.9 (<0.0001)	4.16 (0.04)	4.51 (0.0008)	3.29	0.028	0.2849	0.2565	0.2178	8.42
Resilience	11.2 (<0.0001)	12.2 (<0.0001)	9.06 (0.003)	-	-	16.0 (<0.0001)	6.49	0.016	0.2076	0.1890	0.1513	8.62
Mean cell area	156 (<0.0001)	230 (<0.0001)	9.21 (0.003)	-	-	30.2 (<0.0001)	31.2	148	0.6522	0.6480	0.6424	30.5
Cell density	96.4 (<0.0001)	227 (<0.0001)	5.48 (0.02)	5.21 (0.02)	5.45 (0.02)	72.8 (<0.0001)	21.1	0.46	0.6513	0.6445	0.6337	28.5
Void fraction	144 (<0.0001)	270 (<0.0001)	35.8 (<0.0001)	-	17.9 (<0.0001)	73.9 (<0.0001)	8.92	0.29	0.6919	0.6871	0.6812	35.0

-, Term dropped from equation due to non-significance (*p* > 0.05).

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
