# Peer review of "Optimization of Quality Properties of Gluten-Free Bread by a Mixture Design of Xanthan, Guar, and Hydroxypropyl Methyl Cellulose Gums"

_foods, 2019, doi:10.3390/foods8050156_

Round 1
Reviewer 1 Report
I enjoyed reading the manuscript entitled "optimisation of Quality Properties of Gluten-FreeBread by a Mixture Design of Xanthan, Guar and Hydroxypropyl Methyl Cellulose Gums"; Nevertheless some points in the experimental design are not clear and require more accurate explanation.
1/ the reason behinds the choice of the level addition, where some times the difference is low such as the case of HPMC;
2/ the reason behind the exclusion of references (samples with no additives, samples with one additive by itself)
3/the reasons behind the choice of the mathematical model
4/what about the validation of the model?
please find enclosed a file with more comments

Author Response
Response to Reviewer 1 Comments
Point 1: I enjoyed reading the manuscript entitled "optimisation of Quality Properties of Gluten-Free Bread by a Mixture Design of Xanthan, Guar and Hydroxypropyl Methyl Cellulose Gums"; Nevertheless, some points in the experimental design are not clear and require more accurate explanation.
Response 1: First of all, thank you for the appreciation of our manuscript.
Point 2: the reason behinds the choice of the level addition, where some times the difference is low such as the case of HPMC;
Response 2: Previous breadmaking studies (namely, “Combined effect of xanthan gum on physicochemical and textural properties of gluten-free (GF) batter and bread” / “Physicochemical and textural quality attributes of gluten-free bread formulated with guar gum”] allowed us to define the minimum and maximum level to be evaluated for the three gums. From these studies, we concluded that low doses of xanthan or guar gums (from 0.04 to 0.60%), and high HPMC doses (from 2.80 to 3.60%) were necessary in order to avoid common gluten-free bread flaws. That information is mentioned in lines 156-160.
Point 3: the reason behind the exclusion of references (samples with no additives, samples with one additive by itself)
Response 3: In our experiments, working with a mixture of rice/maize/quinoa flours, with 100% water content, with the use of only one gum did not allowed us to obtain acceptable GF breads, because of the collapsing of batter matrix during proofing or baking. That information is mentioned in lines 156-160
Samples with one additive by itself in the GF breadmaking were evaluated in (Encina-Zelada, C.R.; Cadavez, V.; Monteiro, F.; Teixeira, J.A.; Gonzales-Barron, U. Combined effect of xanthan gum on physicochemical and textural properties of gluten-free batter and bread. Food Res. Int. 2018, 111, 544–555) and in (Encina-Zelada, C.R.; Cadavez, V.; Monteiro, F.; Teixeira, J.A.; Gonzales-Barron, U. Physicochemical and textural quality attributes of gluten-free bread formulated with guar gum. Eur. Food Res. Technol. 2019, 245, 443–458); both papers are mentioned in the References section of the current version of the paper.
Point 4: the reasons behind the choice of the mathematical model
Response 4: Because this study attempts to explain the effect of the dependent variables using simple model responses, linear and two-way interaction terms were deemed sufficient. Although not shown here, cubic or quartic models turned the lack-of-fit values non-significant; however, these higher-order models are devoid of real explanation and, hence, were not considered as suitable in this study. That information is mentioned in lines 243-247.
Point 5: what about the validation of the model?
The authors are aware that all models, including those arising from triangle experiments must be validated. This piece of work comes from an ongoing PhD research, and the validation part has not been included here, to not enlarge further the manuscript which is already very long. However, if the reviewer reckons that the manuscript will not become “too heavy” with more results, the authors can add validation results.
Point 6: please find enclosed a file with more comments
Response 6: Thanks for your observations. All the comments marked in the PDF were addressed and corrected in the newest version of the manuscript.
- The use of comparatives in the Abstract and in the Conclusions were modified, in lines 19-25 and 544-550.
- The chosen triangle design did not allow testing the effect of a three-way interaction; for this, more runs would be needed (i.e., more points within the triangle area). The authors opted for keeping the model simple by studying only quadratic terms and two-way interactions. Three-way ANOVA is of course possible, but only if the triangle experiment would have been designed for that.
Reviewer 2 Report
Manuscript ID: foods-492343
Title: Optimisation of Quality Properties of Gluten-Free Bread by a Mixture
Design of Xanthan, Guar and Hydroxypropyl Methyl Cellulose Gums
Authors: Christian R. Encina-Zelada*, Vasco Cadavez, José A. Teixeira,
Ursula Gonzales-Barron * Submitted to section: Food Quality and Safety.
GENERAL COMENTS:
The manuscript shows very important idea about optimisation of quality properties of gluten-free bread by a mixture of different gums. Very often the interactions between different additives aren’t predictable and should be studied. . Subject is valuable especially for industries. It must be mentioned that Authors put a lot of effort into doing this work Manuscript is well organized but some proposed changes should be done. Some details in introduction, material method and discussion should be completed.
INTRODUCTION SECTION
Introduction section is rather well organized some changes can be done. There should be mentioned more others research about using hydrocolloids in gluten-free bread. Some attempts have been already done to optimize of additives to gluten-free bread. For example quality improvement of gluten-free bread by the addition of some starches/hydrocolloids and their combinations using a definitive screening design was recently done by Bourekoua et al. 2018. Above manuscript should be mentioned in introduction and discussion section. In other research (also not mentioned in manuscript) presented by Demirkeser et al. 2010 different gums (xanthan gum, guar gum, locust bean gum (LBG), hydroxyl propyl methyl cellulose (HPMC), pectin, xanthan-guar, and xanthan-LBG blend) and emulsifiers were used to find the best formulation for gluten-free bread. In other research presented by Mirzaei, & Movahed (2013) the effect of adding k-Carrageenan and Carboxy Methyl Cellulose gums combined with Sodium Stearoyl-2-Lactilate (SSL) on quality of rice bread was studied and should be mentioned.
MATERIALS AND METHODS
Please mention more details about origin of flours (rice, maize, quinoa)(name of the supplier or producer). Please add standard deviations for chemical composition analysis (e.g. protein 7.60±0.42). Please give explanation about recipe and procedure used for gluten-free bread – it was your own recipe and procedure or it was based on recent literature. Please cite the references for texture measurements and parameters calculation or add information that it is your own procedure.
RESULT AND DISCUSSION SECTION
The results are well presented and described in discussion section please mention about proposed literature listed below.
Literature cited above:
Bourekoua, H., Różyło, R., Benatallah, L., Wójtowicz, A., Łysiak, G., Zidoune, M. N., & Sujak, A. (2018). Characteristics of gluten-free bread: quality improvement by the addition of starches/hydrocolloids and their combinations using a definitive screening design. European Food Research and Technology, 244(2), 345–354. https://doi.org/10.1007/s00217-017-2960-9
Cato, L., Gan, J. J., Rafael, L. G. B., & Small, D. M. (2004). Gluten free breads using rice flour and hydrocolloid gums. Food Australia, 56(3), 75–78.
Demirkesen, I., Mert, B., Sumnu, G., & Sahin, S. (2010). Rheological properties of gluten-free bread formulations. Journal of Food Engineering, 96(2), 295–303. https://doi.org/10.1016/j.jfoodeng.2009.08.004
Mirzaei, M., & Movahed, S. (2013). Evaluation of staling rate and quality of gluten-free toast breads on rice flour basis. Research Journal of Applied Sciences, Engineering and Technology, 5(1), 224–227.
Rosell C.M, Rojas J.A, De Barber C.B. (2001) Influence of hydrocolloids on dough rheology and bread quality. Food Hydrocolloid, 15(1), 75–81.
Author Response
Response to Reviewer 2 Comments
Point 1: The manuscript shows very important idea about optimisation of quality properties of gluten-free bread by a mixture of different gums. Very often the interactions between different additives aren’t predictable and should be studied. Subject is valuable especially for industries. It must be mentioned that Authors put a lot of effort into doing this work Manuscript is well organized but some proposed changes should be done. Some details in introduction, material method and discussion should be completed.
Response 1: First of all, thank you for the appreciation of our manuscript.
Point 2: INTRODUCTION SECTION
Introduction section is rather well organized some changes can be done. There should be mentioned more others research about using hydrocolloids in gluten-free bread. Some attempts have been already done to optimize of additives to gluten-free bread. For example, quality improvement of gluten-free bread by the addition of some starches/hydrocolloids and their combinations using a definitive screening design was recently done by Bourekoua et al. 2018. Above manuscript should be mentioned in introduction and discussion section. In other research (also not mentioned in manuscript) presented by Demirkeser et al. 2010 different gums (xanthan gum, guar gum, locust bean gum (LBG), hydroxyl propyl methyl cellulose (HPMC), pectin, xanthan-guar, and xanthan-LBG blend) and emulsifiers were used to find the best formulation for gluten-free bread. In other research presented by Mirzaei, & Movahed (2013) the effect of adding k-Carrageenan and Carboxy Methyl Cellulose gums combined with Sodium Stearoyl-2-Lactilate (SSL) on quality of rice bread was studied and should be mentioned.
Response 2: From the papers that you suggested, we included in the Introduction section information about Bourekoua et al. (2018), Demirkesen et al. (2010) and Mirzaei and Movahe (2013). That information is shown in lines 37-42.
Point 3: MATERIALS AND METHODS
Please mention more details about origin of flours (rice, maize, quinoa) (name of the supplier or producer). Please add standard deviations for chemical composition analysis (e.g. protein 7.60±0.42). Please give explanation about recipe and procedure used for gluten-free bread – it was your own recipe and procedure or it was based on recent literature. Please cite the references for texture measurements and parameters calculation or add information that it is your own procedure.
Response 3: In relation to the Materials and Methods section, we have reformulated it accordingly. More details about flours have been mentioned in lines 66-71. The standard deviations for chemical composition analysis have been added in lines 68-71. The references for texture measurements have been mentioned in line 108 and 130. Explanations about recipe and breadmaking procedure are now displayed in lines 75-87.
Point 4: RESULT AND DISCUSSION SECTION
The results are well presented and described in discussion section please mention about proposed literature listed below.
Response 4: From the papers that you suggested, we included in the Results and Discussion section the outcomes from Bourekoua et al. (2018), Demirkesen et al. (2010), Mirzaei and Movahe (2013). The information is shown in lines 349-354, 419-420 and 472.
Reviewer 3 Report
The manuscript is clearly written and the results, very useful for researchers, would be also of practical application at industrial level.
Few suggestions.
In the Introduction it should be mentioned that the production of gluten free bread without gums is difficult, but not impossible (Pasqualone, A., Caponio, F., Summo, C., Paradiso, V. M., Bottega, G., & Pagani, M. A. (2010). Gluten-free bread making trials from cassava (Manihot esculenta Crantz) flour and sensory evaluation of the final product. International Journal of Food Properties, 13, 562-573).
The rheological properties of batters have been measured (lines 88-100) and mentioned in discussion (lines 239-240 and others), but are not presened in tables. A table showing the rheological properties of batters should be added.
The names of yeasts and microbs should be written in italics.
"ml" ashould be amended to "mL" (lines 77 and 103).
Lines 81-84 are not related to GF breadmaking process, therefore should be moved somewhere else.
Author Response
Response to Reviewer 3 Comments
Point 1: The manuscript is clearly written and the results, very useful for researchers, would be also of practical application at industrial level.
Response 1: First of all, thank you for the appreciation of our manuscript.
Point 2: In the Introduction it should be mentioned that the production of gluten free bread without gums is difficult, but not impossible (Pasqualone, A., Caponio, F., Summo, C., Paradiso, V. M., Bottega, G., & Pagani, M. A. (2010). Gluten-free bread making trials from cassava (Manihot esculenta Crantz) flour and sensory evaluation of the final product. International Journal of Food Properties, 13, 562-573).
Response 2: From the paper that you suggested, we included in the Introduction section information about Pasqualone et al. (2010). That information is shown in lines 41-43.
Point 3: The rheological properties of batters have been measured (lines 88-100) and mentioned in discussion (lines 239-240 and others), but are not presented in tables. A table showing the rheological properties of batters should be added.
Response 3: Done, the rheological (stickiness and firmness) properties of batters are included in Table 2, for all the eleven treatments.
Point 4: The names of yeasts and microbes should be written in italics.
Response 4: Done, in line 75.
Point 5: "ml" should be amended to "mL" (lines 77 and 103).
Response 5: Done, in line 84.
Point 6: Lines 81-84 are not related to GF breadmaking process, therefore should be moved somewhere else.
Response 6: Done, lines 81-84 are now in lines 106-109, in “Rheological properties of GF batter” section.
Round 2
Reviewer 1 Report
The manuscript has been significantly improved.
Author Response
First of all, thanks for your most recent observations about our manuscript. All the comments marked in the Word manuscript by you, were addressed and corrected in the newest version of the manuscript. Please, find enclosed a new Word file including the corrections of your comments.
